# Overexpression of the flagellar motor protein MotB sensitizes *Bacillus subtilis* to aminoglycosides in a motility-independent manner

Mio Uneme, Kazuya Ishikawa, Kazuyuki Furuta, Atsuko Yamashita[ID], Chikara Kaito[ID]*

Graduate School of Medicine, Dentistry and Pharmaceutical Sciences, Okayama University, Okayama, Japan

* ckaito@okayama-u.ac.jp

**Data Availability Statement:** All relevant data are within the manuscript and its Supporting information files.

## Abstract

The flagellar motor proteins, MotA and MotB, form a complex that rotates the flagella by utilizing the proton motive force (PMF) at the bacterial cell membrane. Although PMF affects the susceptibility to aminoglycosides, the effect of flagellar motor proteins on the susceptibility to aminoglycosides has not been investigated. Here, we found that MotB overexpression increased susceptibility to aminoglycosides, such as kanamycin and gentamicin, in *Bacillus subtilis* without affecting swimming motility. MotB overexpression did not affect susceptibility to ribosome-targeting antibiotics other than aminoglycosides, cell wall-targeting antibiotics, DNA synthesis-inhibiting antibiotics, or antibiotics inhibiting RNA synthesis. Meanwhile, MotB overexpression increased the susceptibility to aminoglycosides even in the *motA*-deletion mutant, which lacks swimming motility. Overexpression of the MotB mutant protein carrying an amino acid substitution at the proton-binding site (D24A) resulted in the loss of the enhanced aminoglycoside-sensitive phenotype. These results suggested that MotB overexpression sensitizes *B. subtilis* to aminoglycosides in a motility-independent manner. Notably, the aminoglycoside-sensitive phenotype induced by MotB requires the proton-binding site but not the MotA/MotB complex formation.

## Introduction

The efficacy of antimicrobials is affected by various cellular activities, including drug uptake and efflux, degradation, changes in drug targets, and respiratory pathways. Identifying bacterial factors that affect antibiotic efficacy is important for developing novel drugs that suppress antibiotic-resistant bacteria.

Aminoglycosides are bactericidal antibiotics used against Gram-negative and Gram-positive bacteria. Aminoglycosides target the 30S subunit of the ribosome and exert their bactericidal activity by inhibiting protein synthesis [1,2]. The bactericidal activity of aminoglycosides requires the proton motive force (PMF) of the bacterial cell membrane [3–5]. The reason why PMF is required for the action of aminoglycosides is largely unclear, although it is widely

**Funding:** This study was supported by JSPS Grants-in-Aid for Scientific Research (grants 22K14892, 22H02869, 22K19435, and 23K06130), the Takeda Science Foundation (CK), the Ichiro Kanehara Foundation (CK), the Ryobi Teien Memory Foundation (CK and KI), and Ohmoto Ikueikai Student Grant (RS). The funders had no role in study design, data collection and analysis, decision to publish, or preparation of the manuscript.

**Competing interests:** The authors have declared that no competing interests exist.

accepted that the uptake of positively charged aminoglycosides requires an electric potential difference between the inside and outside of the bacterial cell membrane [4]. Several studies have proposed that the reactive oxygen species produced by aerobic respiration [6] or dysregulation of the electric potential of the bacterial cell membrane [7] affect the bactericidal action of aminoglycosides. In both cases, the PMF of the bacterial cell membrane played an important role in the efficacy of aminoglycosides. Previous studies have revealed that mutations in the electron transport complex [8,9] or F1Fo-ATP synthase [7,10,11], which directly change PMF, alter susceptibility to aminoglycosides or its uptake. However, other bacterial factors that affect the electric potential difference in the bacterial cell membrane and susceptibility to aminoglycosides have not been identified.

Many bacterial species use their flagella to move in liquid or host environments [12]. The bacterial flagellum is a multi-protein complex comprising more than 25 proteins, but its basic structure and rotating mechanism are well conserved across bacterial genera. The bacterial flagellum has three components: basal body, hook, and filament [13]. The basal body is embedded in the bacterial cell membrane and functions as a motor. The hook connects the basal body to the filament and transmits the torque generated by the motor to the filament protruding outside the cell [14]. The driving force of the flagellar motor is the flow of ions through the stator complexes of the flagellar motor. The flagellar stator is a hetero-heptameric membrane protein complex consisting of five A subunits and two B subunits [15,16]. *Escherichia coli* and *Salmonella spp.* have a proton-driven stator, MotA/MotB, whereas *Vibrio spp.* in the oceanic environment have a sodium ion-driven stator, PomA/PomB [17–20]. *Bacillus subtilis* uses both stator complexes, proton-driven MotA/MotB and sodium ion-driven MotP/MotS [YtxD/YtxE], in response to the external environment [21–23].

We hypothesized that proton transfer by MotA/MotB affects bacterial susceptibility to aminoglycosides. Here, we demonstrated that the overexpression of MotB sensitizes *B. subtilis* to aminoglycosides. Unexpectedly, the aminoglycoside-sensitive phenotype induced by MotB did not require the formation of a complex with MotA. Furthermore, MotB overexpression increased the sensitivity to aminoglycosides without affecting *B. subtilis* motility. We propose that overexpressed MotB increases the susceptibility to aminoglycosides in a motility-independent manner. Moreover, we provide an implication for the link between motor proteins and antibiotic susceptibility.

## Results

### Overexpression of MotB makes *B. subtilis* sensitive to aminoglycosides without affecting the motility

We examined the effects of MotA and MotB on the susceptibility of *B. subtilis* to aminoglycosides. The *motA-* and *motB*-deletion mutants showed survival similar to that of the wildtype strain in the presence of kanamycin or gentamicin (Fig 1A). In contrast, the MotB-overexpressed strain displayed decreased survival in the presence of kanamycin or gentamicin compared to the vector-transformed strain but showed similar survival or growth to the vector-transformed strain in the absence of these antibiotics (Fig 1B and S1 Fig). The survival of the MotA-overexpressed strain was similar to that of the vector-transformed strain in the presence of kanamycin or gentamicin (Fig 1B). In the presence of kanamycin, the strain that overexpressed both MotA and MotB showed decreased survival compared to the vector-transformed strain but showed survival similar to that of the MotB-overexpressed strain (Fig 1B). These results suggested that MotB overexpression sensitizes *B. subtilis* to aminoglycosides, but MotA overexpression does not.

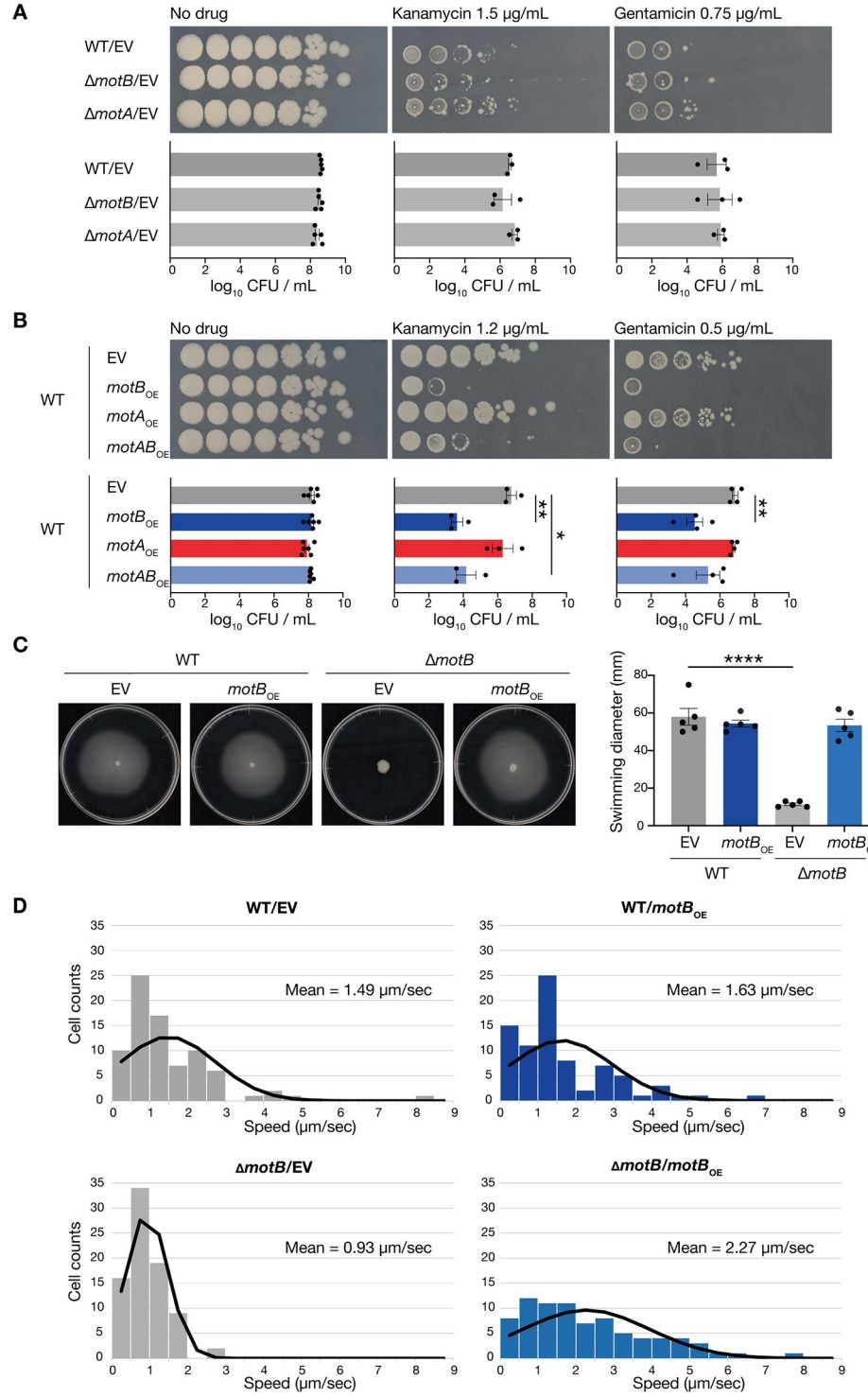

**Fig 1. Overexpression of MotB leads to an aminoglycoside-sensitive phenotype in *B. subtilis* without altering its motility.** A. Overnight cultures of the vector-transformed strain (WT/EV), *motB*-deletion strain (Δ*motB*/EV), and *motA*-deletion strain (Δ*motA*/EV) strain, serially diluted (10-fold) and spotted on LB agar containing 1 mM IPTG with or without kanamycin (1.5 μg/mL) or gentamicin (0.75 μg/mL), and incubated at 37°C. EV means that the strain was transformed using pDR110, an empty vector. Data shown are the means ± standard errors from a minimum of three independent experiments. Statistical analyses were performed using a one-way ANOVA with Dunnett's multiple comparisons test. B. Overnight cultures of the vector-transformed strain (WT/EV), MotB-overexpressed strain (WT/*motB*$_{OE}$), MotA-overexpressed strain (WT/*motA*$_{OE}$), and MotA/MotB-overexpressed strain (WT/*motAB*$_{OE}$), serially

diluted (10-fold) and spotted on LB agar containing 1 mM IPTG with or without kanamycin (1.2 µg/mL) or gentamicin (0.5 µg/mL), and incubated at 37˚C. Data shown are the means ± standard errors from a minimum of three independent experiments. Statistical analyses were performed using a one-way ANOVA with Dunnett's multiple comparisons test. *, $p < 0.05$; **, $p < 0.01$. C. The swimming motility of the vector-transformed strain (WT/EV), MotB-overexpressed strain (WT/$motB_{OE}$), *motB*-deletion strain (Δ*motB*/EV), and *motB*-deletion strain overexpressing MotB (Δ*motB*/$motB_{OE}$) on 0.3% soft agar medium containing 1 mM IPTG. Data shown are the means ± standard errors from five independent experiments. Statistical analyses were performed using a one-way ANOVA with Dunnett's multiple comparisons test. ****, $p < 0.0001$. D. Histograms and normal distribution curves of the single-cell speeds of the vector-transformed strain (WT/EV), MotB-overexpressed strain (WT/$motB_{OE}$), *motB*-deletion strain (Δ*motB*/EV), and *motB*-deletion strain overexpressing MotB (Δ*motB*/$motB_{OE}$) in LB medium with 1 mM IPTG. Δ*motB*/EV was significantly different from WT/EV and Δ*motB*/$motB_{OE}$ ($p < 0.05$), but WT/$motB_{OE}$ was not significantly different from WT/EV ($p = 0.73$). Statistical analyses were performed using Kolmogorov-Smirnov tests.

Next, we examined if MotB overexpression affects *B. subtilis* motility. To evaluate the motility of each strain, we observed spreading in semi-solid agar plates. Notably, there was no significant difference in the motility between the vector-transformed strain and the MotB-overexpressed strain (Fig 1C). In these assay conditions, the *motB*-deletion mutant showed decreased motility compared to the wildtype strain, and motility was restored by MotB overexpression (Fig 1C). To compare the motility of each strain more directly, we measured the swimming speed of single cells using a microscope. There was no significant difference in the swimming speed between the vector-transformed strain and the MotB-overexpressed strain (Fig 1D). The swimming speed of the *motB*-deletion mutant was decreased compared to the wildtype strain, and the speed was increased by MotB overexpression to a similar extent as the MotB-overexpressed wildtype strain (Fig 1D). Overexpression of MotB in the wildtype strain did not affect the swimming speed but made a difference in the susceptibility to aminoglycosides, suggesting no correspondence between the bacterial swimming speed and the aminoglycoside-sensitive phenotype. Taken together, these results indicate that MotB overexpression leads to an aminoglycoside-sensitive phenotype without affecting motility.

## Overexpression of MotB sensitizes *B. subtilis* to aminoglycosides in the absence of MotA

MotB forms a complex with MotA and functions as a flagellar motor [24]. We examined the requirement of MotA for the aminoglycoside-sensitive phenotype caused by MotB overexpression. Overexpression of MotB in the *motA*-deletion mutant resulted in decreased survival compared to that in the vector-transformed *motA*-deletion mutant in the presence of kanamycin or gentamicin (Fig 2). This result suggests that MotA is not required for the aminoglycoside-sensitive phenotype caused by MotB overexpression.

## Overexpression of MotB sensitizes *B. subtilis* only to aminoglycosides

We examined whether MotB overexpression affects *B. subtilis* susceptibility to antimicrobials other than aminoglycosides. The survival of the MotB-overexpressed strain was similar to that of the vector-transformed strain in the presence of ribosome-targeting antibiotics other than aminoglycosides, including erythromycin, chloramphenicol, and tetracycline (Fig 3). In addition, the MotB-overexpressed strain showed survival similar to that of the vector-transformed strain in the presence of cell wall-targeting antibiotics, including vancomycin and ampicillin, DNA synthesis-inhibiting antibiotics, including ciprofloxacin and levofloxacin, and an RNA synthesis-inhibiting antibiotic, rifampicin (Fig 3). The minimum inhibitory concentrations (MICs) for kanamycin and gentamicin of the MotB-overexpressed strain decreased

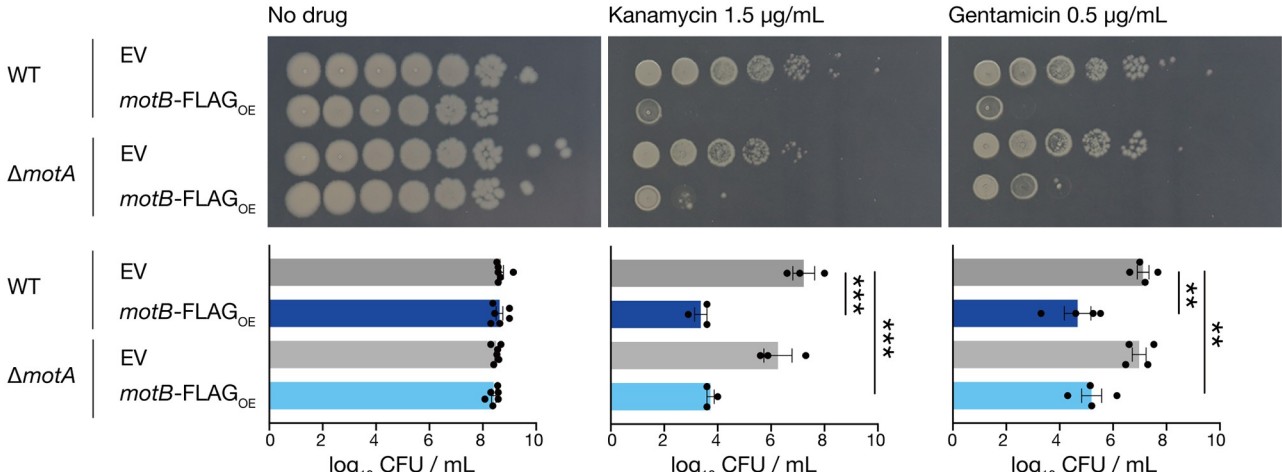

**Fig 2. Overexpression of MotB causes the aminoglycoside-sensitive phenotype independent of MotA.** Overnight cultures of the vector-transformed strain (WT/EV), MotB-overexpressed strain (WT/*motB*-FLAG$_{OE}$), *motA*-deletion strain ($\Delta motA$/EV), and MotB-overexpressing *motA*-deletion strain ($\Delta motA$/*motB*-FLAG$_{OE}$), serially diluted (10-fold) and spotted on LB agar containing 1 mM IPTG with or without kanamycin (1.5 µg/mL) or gentamicin (0.5 µg/mL), and incubated at 37˚C. Data shown are the means ± standard errors from a minimum of three independent experiments. Statistical analyses were performed using a one-way ANOVA with Dunnett's multiple comparisons test. **, $p < 0.01$; ***, $p < 0.001$.

approximately 1.5-fold compared with those of the vector-transformed strain, but the MICs for the other drugs were the same between the MotB-overexpressed strain and the vector-transformed strain (Table 1). These results suggest that MotB overexpression does not affect susceptibility to antimicrobials other than aminoglycosides and leads to sensitive phenotypes specific to aminoglycosides.

## Overexpression of MotS, a homolog of MotB, does not sensitize *B. subtilis* to aminoglycosides

In *B. subtilis*, MotP/MotS converts the transmembrane gradient of sodium ions into energy for flagellar rotation in response to environmental changes, such as salinity changes [21–23]. Since MotS is homologous to MotB [21], we examined whether MotS overexpression sensitizes *B. subtilis* to aminoglycosides. The survival of the MotS-overexpressed strain was similar to that of the vector-transformed strain in the presence of kanamycin or gentamicin (Fig 4A). In addition, the strain overexpressing MotS and MotP, the homolog of MotA, showed survival similar to that of the vector-transformed strain in the presence of kanamycin or gentamicin (Fig 4A).

To confirm the expression of MotS and MotB, a FLAG tag sequence was fused to the C-terminus of *motS* or *motB*. Western blot analyses confirmed MotS-FLAG and MotB-FLAG expressions (Fig 4B, S1 Raw images). These results suggest that the specific function of MotB, which does not exist in MotS, leads to susceptibility to aminoglycosides.

## The proton-binding site of MotB is important for the aminoglycoside-sensitive phenotype

When MotA/MotB or MotP/MotS function as a complex, MotB or MotS have been shown to be the major determinants of ion bindings [25]. Based on the finding that overexpression of MotS, which binds sodium ions, did not cause an aminoglycoside-sensitive phenotype, we examined whether the proton-binding site of MotB plays a role in the aminoglycoside-

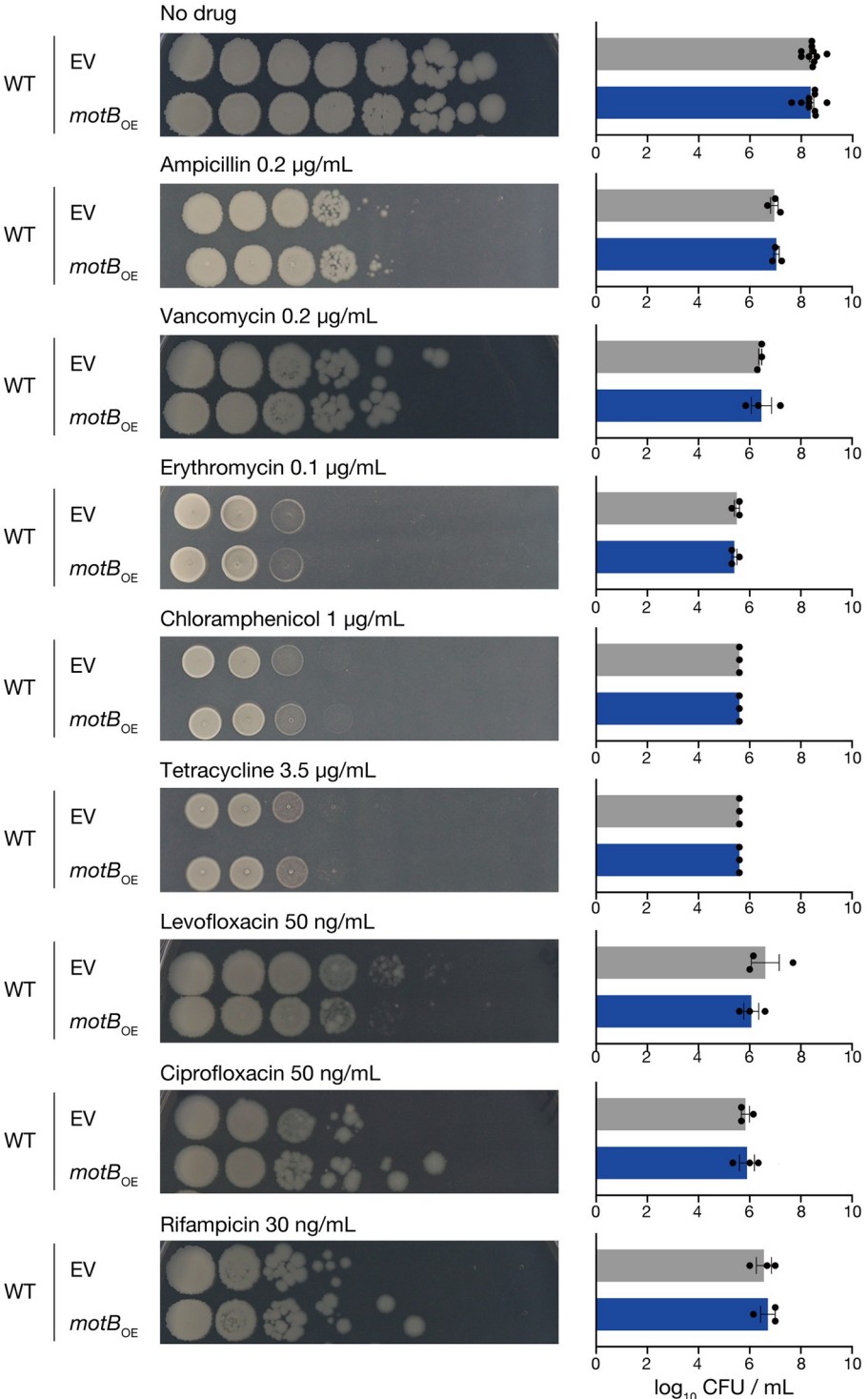

**Fig 3. Overexpression of MotB does not alter *B. subtilis* susceptibility to antibiotics other than aminoglycosides.** Overnight cultures of the vector-transformed strain (WT/EV) and the MotB-overexpressed strain (WT/*motB*$_{OE}$), serially diluted (10-fold) and spotted on LB agar medium containing 1 mM IPTG with or without ampicillin (0.2 μg/mL), vancomycin (0.2 μg/mL), erythromycin (0.1 μg/mL), chloramphenicol (1 μg/mL), tetracycline (3.5 μg/mL), levofloxacin (50 ng/mL), ciprofloxacin (50 ng/mL), or rifampicin (30 ng/mL), and incubated at 37°C. Data shown are the means ± standard errors from a minimum of three independent experiments. Statistical analyses were calculated using a one-way ANOVA with Dunnett's multiple comparisons test.

**Table 1. MIC values of the MotB-overexpressed strain against various antibiotics.**

| Strain | MIC (µg/mL) | | | | | | | | | |
|---|---|---|---|---|---|---|---|---|---|---|
| | Kan | Gen | Amp | Van | Erm | Crm | Tet | Levo | Cip | Rif |
| EV | 3.0 | 1.0 | 0.30 | 0.30 | 0.094 | 2.3 | 4.5 | 0.075 | 0.075 | 0.068 |
| *motB*_OE | 1.8 | 0.68 | 0.30 | 0.30 | 0.094 | 2.3 | 4.5 | 0.075 | 0.075 | 0.068 |

Kan, Kanamycin; Gen, Gentamicin; Amp, Ampicillin; Van, Vancomycin; Erm, Erythromycin; Crm, Chloramphenicol; Tet, Tetracycline; Levo, Levofloxacin; Cip, Ciprofloxacin; Rif, Rifampicin. Data are an average from three replicates.

sensitive phenotype. Based on reports that Asp24 of MotB electrically interacts with protons to achieve proton transport in *B. subtilis* [26,27], we generated *motB* mutant genes with Asp24 substituted with Ala (D24A) or Glu (D24E). Based on the amino acid characteristics, D24A MotB does not interact with protons, whereas D24E MotB mimics the interaction of wildtype MotB with protons [27]. All *motB* genes were fused to a FLAG tag sequence at the C-terminus to examine the localization of the overexpressed MotB. The D24A MotB-

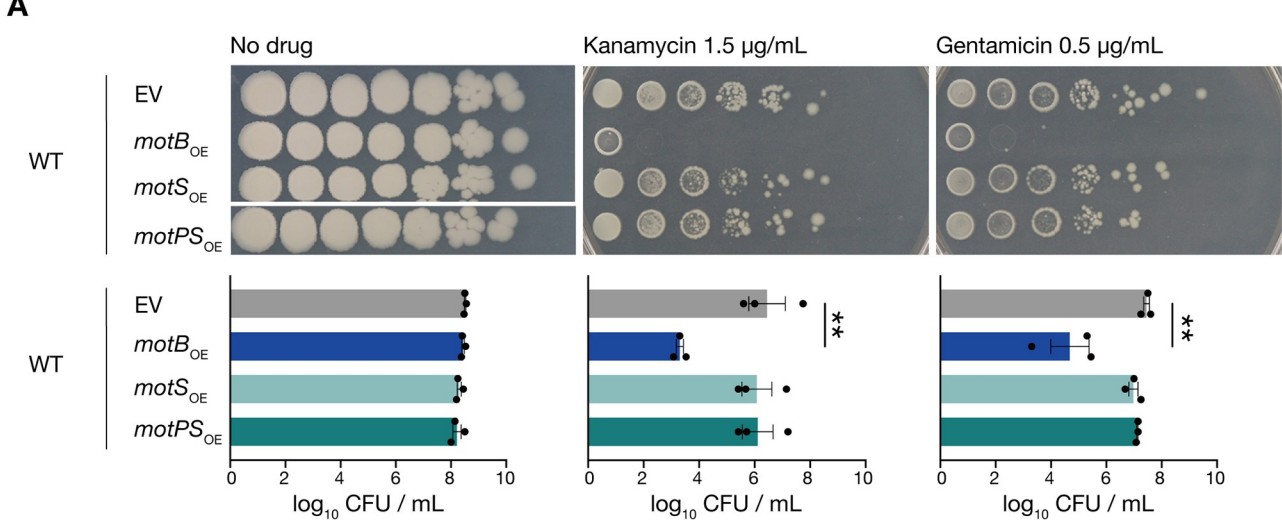

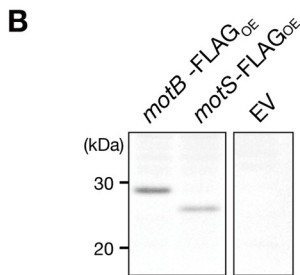

**Fig 4. Overexpression of MotS, a homolog of MotB, does not cause the aminoglycoside-sensitive phenotype.** A. Overnight cultures of the vector-transformed strain (WT/EV), MotB-overexpressed strain (WT/*motB*_OE), MotS-overexpressed strain (WT/*motS*_OE), and MotP/MotS-overexpressed strain (WT/*motPS*_OE), serially diluted (10-fold) and spotted on LB agar containing 1 mM IPTG with or without kanamycin (1.5 µg/mL) or gentamicin (0.5 µg/mL), and incubated at 37˚C. Data shown are the means ± standard errors from three independent experiments. Statistical analyses were calculated using a one-way ANOVA with Dunnett's multiple comparisons test. **, $p < 0.01$. B. Western blot using an anti-FLAG antibody for the whole cell fraction extracted from overnight cultures of the vector-transformed strain (EV), MotB-overexpressed strain (*motB*-FLAG_OE), and MotS-overexpressed strain (*motS*-FLAG_OE).

overexpressed strain did not show significantly decreased survival compared with the vector-transformed strain in the presence of kanamycin or gentamicin (Fig 5A). In contrast, in the presence of kanamycin, the D24E MotB-overexpressed strain showed less survival than the vector-transformed strain (Fig 5A). Next, to examine the possibility that the amino acid substitutions in MotB affect subcellular localization, cytosolic and membrane fractions were prepared from these strains, and wildtype and mutant MotB proteins were detected by western blotting. FoF1-ATP synthase subunit a (AtpB; [28]) and an RNA polymerase sigma factor (SigB; [29,30]) were used as marker proteins for the membrane fraction and the cytosolic fraction, respectively. Wildtype MotB, D24A MotB, and D24E MotB were not detected in the cytosolic fraction in which the cytosolic protein SigB was detected (Fig 5B, S2 Raw images). These proteins were only detected in the membrane fraction in which the membrane protein AtpB was detected (Fig 5B, S2 Raw images). The expression of D24A MotB was higher than that of wildtype MotB, whereas that of D24E MotB was comparable to that of wildtype MotB (Fig 5B, S2 Raw images). These results suggest that D24A MotB loses its activity, whereas D24E MotB retains it, leading to an aminoglycoside-sensitive phenotype. These findings suggest that the proton-binding site of MotB plays an important role in the aminoglycoside-sensitive phenotype.

The increased expression of the alanine-substituted MotB at the conserved Asp residue has also been reported in *E. coli* [26]. We speculate that D24A MotB was more abundant than wildtype MotB due to the structural stabilization of the MotA/MotB complex because the alanine substitution causes structural alteration of *E. coli* MotA/MotB [31].

## Determination of MotB expression level causing aminoglycoside sensitivity

To determine the protein expression level of MotB that causes the motility activity and the aminoglycoside-sensitive phenotype, we examined the phenotypes of the *motB*-deletion mutant transformed with *motB*-FLAG, which was cultured in the presence of various concentrations of isopropyl β-D-thiogalactopyranoside (IPTG). Addition of 0.05 mM IPTG restored the motility of the *motB*-deletion mutant to the level of the wildtype strain (Fig 6A), but it did not cause the aminoglycoside sensitive-phenotype (Fig 6B). Addition of 0.1 mM IPTG restored the motility activity and caused the aminoglycoside-sensitive phenotype (Fig 6A and 6B). Thus, a higher concentration of IPTG (0.1 mM) is required to cause the aminoglycoside-sensitive phenotype than to cause the motility phenotype (Fig 6C). Western blot analysis revealed that the protein expression level of MotB was greater in 0.1 mM IPTG than in 0.05 mM IPTG (Fig 6D, S3 Raw images). These results suggest that larger expression levels of MotB are required to cause the aminoglycoside-sensitive phenotype than to cause the motility activity.

To understand the MotB complexes that cause the aminoglycoside-sensitive phenotype, we performed a blue native polyacrylamide gel electrophoresis and the subsequent western blot analysis. With 0.1 mM IPTG, we detected protein bands with various sizes of 60 kDa (black arrowhead) and larger than 60 kDa (Fig 6E, S3 Raw images). The signal intensities of the 60 kDa band were comparable between 0.1 mM and 0.05 mM IPTG conditions (Fig 6E, S3 Raw images). The signal intensities of bands larger than 60 kDa were greater with 0.1 mM IPTG than with 0.05 mM IPTG (Fig 6E, S3 Raw images). We estimate that the 60 kDa band represents homodimers of MotB-FLAG because MotB has been reported to form a dimer [32], and the theoretical molecular size of the dimer of MotB-FLAG is 61 kDa. The bands with larger molecular weights may contain the MotA/MotB complex, the complex with flagellar motor, or other unknown complexes. These results suggest that overexpressed MotB exists as protein complexes to cause the aminoglycoside-sensitive phenotype.

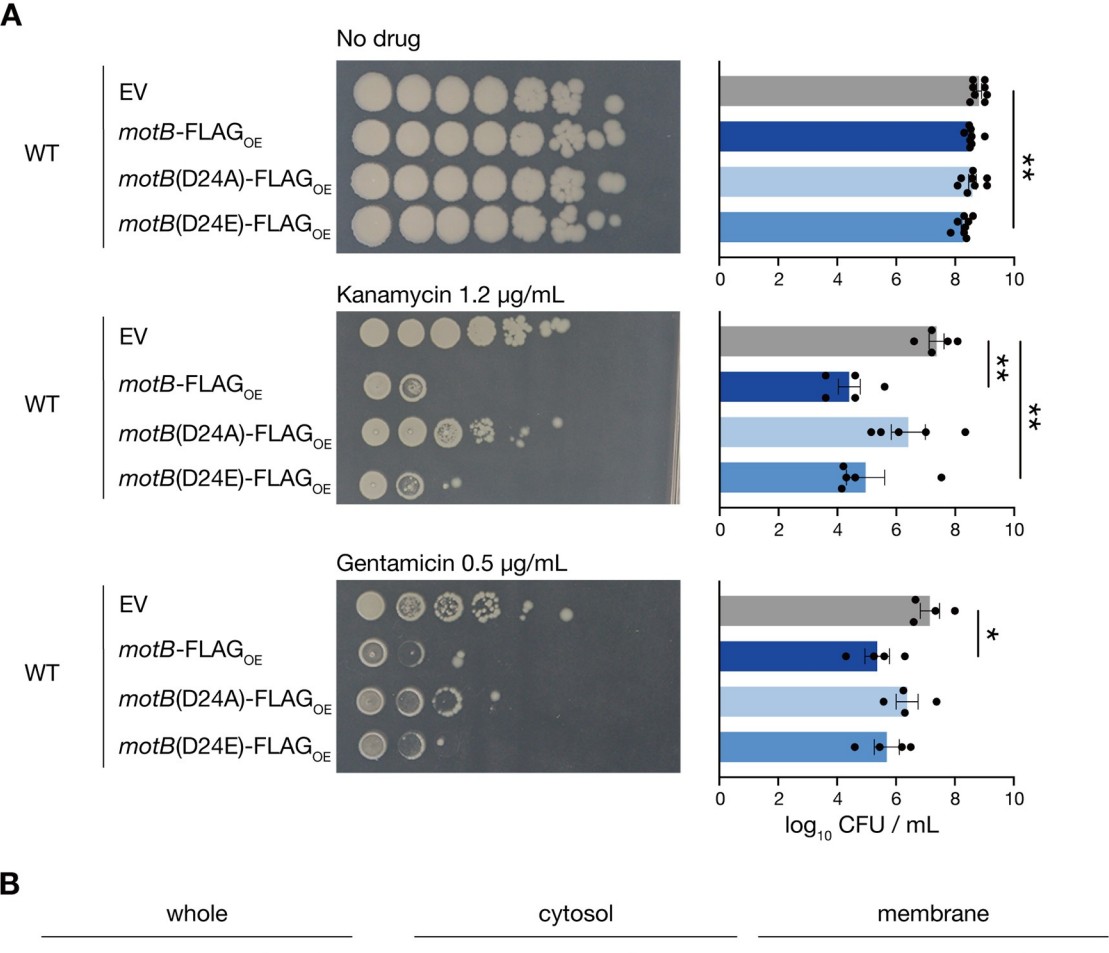

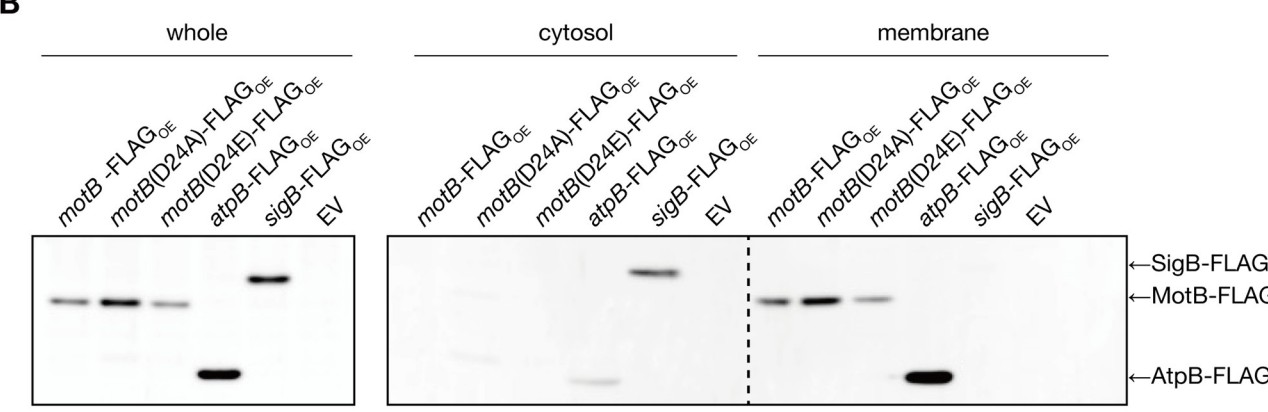

**Fig 5. The proton-binding site of MotB is required for aminoglycoside sensitization.** A. Overnight cultures of the vector-transformed strain (WT/EV), MotB-overexpressed strain (WT/*motB*-FLAG_{OE}), D24A MotB-overexpressed strain (WT/*motB*(D24A)-FLAG_{OE}), and D24E MotB-overexpressed strain (WT/*motB*(D24E)-FLAG_{OE}), serially diluted (10-fold) and spotted on LB agar containing 1 mM IPTG with or without kanamycin (1.2 µg/mL) or gentamicin (0.5 µg/mL), and incubated at 37°C. Data shown are the means ± standard errors from a minimum of four independent experiments. Statistical analyses were performed using a one-way ANOVA with Dunnett's multiple comparisons test. *, $p < 0.05$; **, $p < 0.01$. B. Western blot using an anti-FLAG antibody for the whole cell fraction, membrane fraction, and cytosolic fraction extracted from overnight cultures of the vector-transformed strain (WT/EV), MotB-overexpressed strain (WT/*motB*-FLAG_{OE}), D24A MotB-overexpressed strain (WT/*motB*(D24A)-FLAG_{OE}), D24E MotB-overexpressed strain (WT/*motB*(D24E)-FLAG_{OE}), AtpB-overexpressed strain (WT/*atpB*-FLAG_{OE}), and SigB-overexpressed strain (WT/*sigB*-FLAG_{OE}).

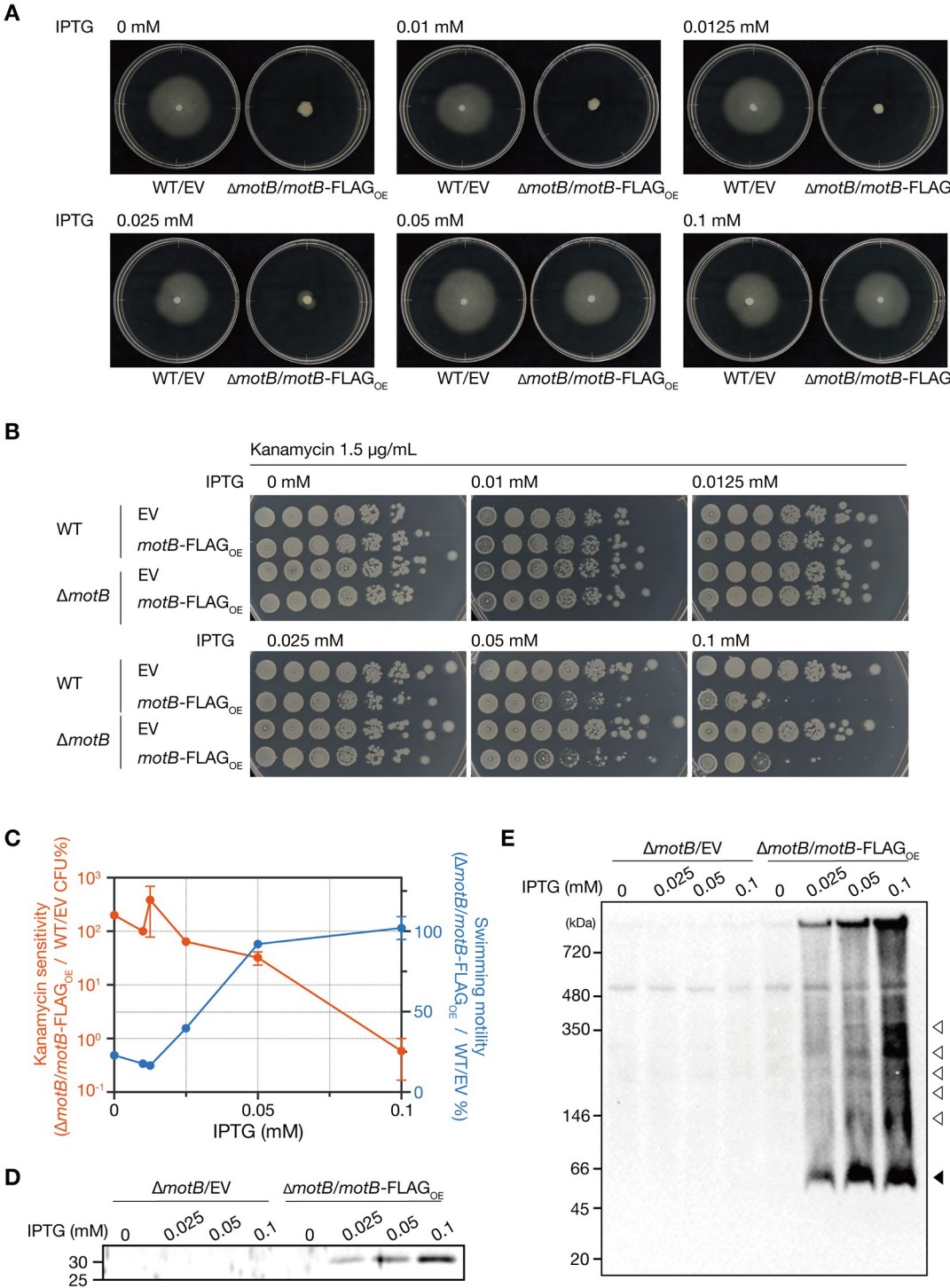

**Fig 6. The overexpressed MotB exists as large protein complexes.** A. Swimming motility of the vector-transformed strain (WT/EV) and *motB*-deletion strain overexpressing MotB (Δ*motB*/*motB*-FLAG$_{OE}$) on 0.3% soft agar medium containing various concentrations of IPTG. B. Overnight cultures of the vector-transformed strain (WT/EV), MotB-overexpressed strain (WT/*motB*-FLAG$_{OE}$), *motB*-deletion strain (Δ*motB*/EV), and *motB*-deletion strain overexpressing MotB (Δ*motB*/*motB*-FLAG$_{OE}$), serially diluted (10-fold) and spotted on LB agar containing various concentrations of IPTG with kanamycin (1.5 μg/mL), and incubated at 37°C. C. Relationship between IPTG concentration, swimming motility, and kanamycin sensitivity in the *motB*-deletion strain overexpressing MotB. The CFU ratio of the *motB*-deletion strain overexpressing MotB (Δ*motB*/*motB*-FLAG$_{OE}$) to the vector-transformed strain (EV) in the presence of kanamycin is shown in the left vertical axis (orange symbols), and the

swimming ratio of the *motB*-deletion strain overexpressing MotB (Δ*motB*/*motB*-FLAG$_{OE}$) to the vector-transformed strain (EV) is shown in the right vertical axis (blue symbols). Data shown are the average and range of two independent experiments including the results shown in A and B. D. Western blot for the whole cell fraction extracted from the *motB*-deletion strain (Δ*motB*/EV) and the *motB*-deletion strain overexpressing MotB (Δ*motB*/*motB*-FLAG$_{OE}$) that were cultured in the presence of various concentrations of IPTG. E. The membrane fractions extracted from the *motB*-deletion strain (Δ*motB*/EV) and *motB*-deletion strain overexpressing MotB (Δ*motB*/*motB*-FLAG$_{OE}$) that were cultured in the presence of various concentrations of IPTG were solubilized with 0.5% n-dodesyl-β-D-maltoside and subjected to blue native polyacrylamide gel electrophoresis and western blot. A black arrowhead indicates a 60 kDa band, and white arrowheads indicate 146, 180, 220, 180, and 380 kDa bands.

## Discussion

In this study, we found that the overexpression of MotB, a flagellar motor protein, sensitizes *B. subtilis* to aminoglycosides. Notably, we show that MotB leads to aminoglycoside susceptibility in a motility-independent manner. Overexpression of MotB in the *motA*-deletion mutant led to an aminoglycoside-sensitive phenotype, indicating that MotB induces aminoglycoside susceptibility independent of MotA. This is the first study to suggest that MotB has potential other than as a flagellar motor protein.

Overexpression of MotB did not affect *B. subtilis* susceptibilities to ribosome-targeting antibiotics (erythromycin, chloramphenicol, and tetracycline), DNA synthesis-inhibiting antibiotics (levofloxacin and ciprofloxacin), RNA synthesis-inhibiting antibiotics (rifampicin), or cell wall-targeting antibiotics (ampicillin and vancomycin; Fig 3). The aminoglycosides, which are affected by MotB overexpression exclusively regarding efficacy, have a structure containing an aminocyclitol ring (streptidine or 2-deoxystreptamine) and two or more amino sugars linked by glycosidic bonds to it [33]. They contain abundant basic ionizable amino groups as the substituents, which make them positively charged and strongly polar compounds [34,35] (S2 Fig). Therefore, it has been suggested that the importance of PMF is inducing their crossing of the cell membrane and uptake into the bacterial cell. Vancomycin and ampicillin target cell wall synthesis, and their uptake into bacterial cells is not required for antibiotic action [36,37]. In addition, erythromycin, chloramphenicol, tetracycline, levofloxacin, ciprofloxacin, and rifampicin are hydrophobic molecules, and their uptake into bacterial cells is unlikely to be as affected by membrane potential as aminoglycosides. Since MotB overexpression leads to susceptibility specific to aminoglycosides among the antibiotics examined in this study, MotB overexpression may change cellular activities, such as PMF, which specifically affects aminoglycoside susceptibility.

This study revealed that an amino acid substitution in the proton-binding site of MotB (D24A) affects its ability to cause an aminoglycoside-sensitive phenotype. In addition, MotB overexpression resulted in an aminoglycoside-sensitive phenotype in the *motA*-deletion mutant. These findings suggest that interactions MotB with MotA are not required for the phenotype but that its proton-binding site plays an important role in inducing the aminoglycoside-sensitive phenotype. We considered the possibility that MotB overexpression may change PMF by interacting with protons; however, it is difficult to imagine that MotB can bind protons and alter PMF without forming a complex with MotA. In fact, a single-cell motility assay revealed that MotB overexpression did not affect the swimming speed. Previous studies have analyzed various MotB mutants [38,39], but no studies have analyzed the function of MotB in the absence of MotA. This study revealed that overexpressed MotB exists as protein complexes. Future studies are needed to examine the effect of MotB overexpression on PMF or complex formation and the molecular mechanisms by which MotB overexpression causes the aminoglycoside-sensitive phenotype. Furthermore, in addition to the proton binding site in the transmembrane domain, MotB has the peptide glycan binding motif and the plug region,

which have important roles in the regulation of proton translocation [39,40]. The requirement of these regions of MotB for inducing the aminoglycoside-sensitive phenotype should be investigated in the future.

MotB is well-conserved among various flagellated bacteria. These findings suggest that drugs that increase MotB expression have the potential to increase aminoglycoside efficacy. The molecular mechanisms that regulate *motAB* expression have been studied in several bacterial species. In *Dickeya dadantii*, SlyA regulates MotA and MotB *expression via* the PhoP-PhoQ two-component system [41]. In *Listeria monocytogenes*, *motAB* expression is regulated by the virulence regulator PrfA [42]. In *Rhizobium leguminosarum*, VisN/R-Rem regulates the expression of *motAB* [43]. This information regarding the regulation of *motAB* is important to establish research strategies for identifying drugs that activate *motAB* transcription. Notably, aminoglycosides have been in use since the 1940s. Drugs that increase the efficacy of aminoglycosides could be useful in combating aminoglycoside-resistant bacteria.

## Materials and methods

### Bacteria and culture conditions

*B. subtilis* 168 *trpC2* and the mutant strains were aerobically cultured in LB at 37°C. *B. subtilis* gene deletion mutants carrying erythromycin-resistant gene or *B. subtilis* strain transformed with pDR110 were cultured on an LB agar plate supplemented with erythromycin (1 μg/mL) or spectinomycin (100 μg/mL), respectively. To overexpress MotB, bacterial strains were cultured in the LB medium supplemented with 1 mM IPTG. The bacterial strains and plasmids used in this study are listed in Table 2.

### Gene deletion mutant

Genomic DNA was extracted from the *motB*-deletion mutant (BSU13680) or the *motA*-deletion mutant (BSU13690) [44] by using the QIAamp DNA Blood Mini Kit (Qiagen). It was

**Table 2. List of bacterial strains and plasmids used.**

| Strain or Plasmid | Genotype or characteristics | Reference |
|---|---|---|
| Strains | | |
| *Bacillus subtilis* 168 | *trpC2* | BGSC |
| BSU13680 | *trpC2 ΔmotB*; Erm^r | [44] |
| BSU13690 | *trpC2 ΔmotA*; Erm^r | [44] |
| BSU13680-ML | *trpC2 ΔmotB*; markerless | This study |
| BSU13690-ML | *trpC2 ΔmotA*; markerless | This study |
| Plasmids | | |
| pDR110 | An *amyE* integration vector, Amp^r, Spc^r | BGSC [44] |
| pDR244 | Cre recombinase-expressing plasmid, Amp^r, Spc^r | BGSC [44] |
| pDR110-motB | pDR110 with *motB*, Amp^r, Spc^r | This study |
| pDR110-motA | pDR110 with *motA*, Amp^r, Spc^r | This study |
| pDR110-motAB | pDR110 with *motAB*, Amp^r, Spc^r | This study |
| pDR110-motS | pDR110 with *motS*, Amp^r, Spc^r | This study |
| pDR110-motPS | pDR110 with *motPS*, Amp^r, Spc^r | This study |
| pDR110-motB-FLAG | pDR110 with *motB*-FLAG, Amp^r, Spc^r | This study |
| pDR110-motS-FLAG | pDR110 with *motS*-FLAG, Amp^r, Spc^r | This study |
| pDR110-atpB-FLAG | pDR110 with *atpB*-FLAG, Amp^r, Spc^r | This study |
| pDR110-sigB-FLAG | pDR110 with *sigB*-FLAG, Amp^r, Spc^r | This study |
| pDR110-motB(D24A)-FLAG | pDR110 with *motB*(D24A), Amp^r, Spc^r | This study |
| pDR110-motB(D24E)-FLAG | pDR110 with *motB*(D24E), Amp^r, Spc^r | This study |

used to transform *B. subtilis* 168 *trpC2* strain [45]. Erythromycin-resistant colonies were obtained on LB plates supplemented with erythromycin (1μg/mL). The erythromycin-resistant cassette in the mutant strains was excised using pDR244, a temperature-sensitive plasmid expressing a Cre recombinase. The desired deletion of *motB* or *motA* was confirmed using PCR.

### Plasmid construction

DNA fragments containing *motB*, *motA*, *motAB*, *motS*, *motP*, *atpB*, and *sigB* were amplified by PCR using specific primer pairs (Table 3) and a genomic DNA from 168 *trpC2* as a template. The amplified DNA fragments were inserted into the multiple cloning site of pDR110,

**Table 3. Primers used in this study.**

| | |
|---|---|
| Primers to construct pDR110-motB and to confirm the *motB* integration | |
| motB_F_SalI | GTCGTCGACGGACAAGCACCGAAAGTCAT |
| motB_R_SphI | GCAGCATGCAGGTAGAGATGTGCACCGAAA |
| Primers to construct pDR110-motA and to confirm the *motA* integration | |
| motA_F_SalI | GTCGTCGACGGGCACCAAAACCGATATTA |
| motA_R_SphI | GCAGCATGCTCGCGTACAGCACAATAA |
| Primers to construct pDR110-motAB and to confirm the *motAB* integration | |
| motA_F_SalI | GTCGTCGACGGGCACCAAAACCGATATTA |
| motB_R_SphI | GCAGCATGCAGGTAGAGATGTGCACCGAAA |
| Primers to construct pDR110-motS and to confirm the *motS* integration | |
| motS_F_NheI | GCTGCTAGCTTTCAGCTCACGGGAAGAAT |
| motS_R_SphI | GCAGCATGCGATACGTTCTGAACAGC |
| Primers to construct pDR110-motPS and to confirm the *motPS* integration | |
| motP_F_SalI | GTCGTCGACGAGCAAGCTTCACCTTTATGG |
| motS_R_SphI | GCAGCATGCGATACGTTCTGAACAGC |
| Primers to amplify *motB*-FLAG | |
| motB-Flag_F | TAGCAAAAAAGGAAGCCTTGTGA |
| motB-Flag_R | CTTGTCATCGTCGTCCTTGTAGTCTTTTTCATTTGTTTCCGCTG |
| Primers to amplify *motS*-FLAG | |
| motS-Flag_F | TAAAGGTCGTTTTTTACTTTGTT |
| motS-Flag_R | CTTGTCATCGTCGTCCTTGTAGTCCGAAGAGGTCGTTTTTGATT |
| Primers to amplify *atpB*-FLAG | |
| atpB_F | GTCGTCGACCAGCTTAAACGTTCATCAATGG |
| atpB_R_FLAG | GCAGCATGCTTATTTATCATCATCATCTTTATAATCATGATCATGACTGATTTTATGAGACAT |
| Primers to amplify *sigB*-FLAG | |
| sigB-SalI-F | GTCGTCGACCGCTCATGGATGAAGTCAGA |
| sigB-flag-SphI-R | GCAGCATGCTTATTTATCATCATCATCTTTATAATCCATTAACTCCATCGAGGGATCT |
| Primers to amplify *motB* (D24A)-FLAG | |
| motB(D24A)_F | ATGGCTCGTTCCTTACGCCGCAATCCTTACTCTTCTCCTGGC |
| motB(D24A)_R | GCCAGGAGAAGAGTAAGGATTGCGGCGTAAGGAACGAGCCAT |
| Primers to amplify *motB* (D24E)-FLAG | |
| motB(D24E)_F | ATGGCTCGTTCCTTACGCCGAAATCCTTACTCTTCTCCTGGC |
| motB(D24E)_R | GCCAGGAGAAGAGTAAGGATTTCGGCGTAAGGAACGAGCCAT |
| Primers to confirm the replacement of *amyE* locus | |
| amyE_F | TACAGCACCGTCGATCAAAA |
| amyE_R | CTCGGTCCTCGTTACACCAT |

resulting in the plasmids pDR110-motB, pDR110-motA, pDR110-motAB, pDR110-motP, pDR110-motS, pDR110-motPS, pDR110-atpB-FLAG, and pDR110-sigB-FLAG.

The FLAG tag sequence was added to pDR110-motB or pDR110-motS by inverse PCR using specific primer pairs (Table 3) and pDR110-motB or pDR110-motS as a template. The desired addition of a FLAG tag at the C-terminus of MotB or MotS was confirmed by Sanger sequencing. To introduce amino acid substitutions (D24A and D24E) into MotB, thermal cycling reactions were performed using primer pairs (Table 3), pDR110-motB-FLAG as a template, and KOD Plus DNA polymerase (Toyobo). The product of the thermal cycling reaction was digested with DpnI and used to transform *E. coli* JM109. The plasmid was extracted, and the desired mutations were confirmed by Sanger sequencing.

## Transformation

Natural transformation was performed to transform 168 *trpC2* with the plasmids [45]. The colonies were selected on LB plates supplemented with spectinomycin (100 μg/mL). The desired recombination of the plasmid into the *amyE* locus was confirmed by PCR using primer pairs (Table 3) and genomic DNA as the template.

## Evaluation of antibiotic susceptibility

The autoclaved LB agar medium was mixed with 1 mM IPTG and antimicrobials and poured into round or square dishes. *B. subtilis* overnight cultures were serially diluted 10-fold with LB broth in a 96-well microplate, and 5 μL of the diluted bacterial solutions were spotted onto LB plates with or without antibiotics using an 8-channel Pipetman. The plates were incubated overnight at 37˚C. Colonies were photographed using a digital camera and counted.

The criteria for CFU counting were set as follows. i) When 5 or more colonies were observed in the most diluted spot, we counted the number of colonies. ii) When 4 or fewer colonies were observed in the most diluted spot, we counted the number of colonies in the next most diluted spot. iii) When the number of colonies could not be counted due to crowded colonies, we considered the number of colonies to be 20. iv) When the colonies were observed only at the edge of the spot, like a circle, we considered the number of colonies to be 10.

## Evaluation of MIC

Overnight cultures in the presence of 1 mM IPTG were diluted to $10^4$ CFU/5 μL and spotted on LB agar plates with 1 mM IPTG and serial 1.5-fold dilutions of antibiotics. The MIC was determined as the minimum concentration at which no colonies were observed. The values reported are the mean of three replicates.

## Swimming motility assay

2 μL of an $OD_{600}$-adjusted overnight culture was inoculated at the center of LB plates [0.3% (W/V) agar] containing 1 mM IPTG. After incubation at 37˚C for 24 h, the diameter of the bacterial spread was measured.

## Single-cell motility assay

For observation of single-cell motility, a previously described method [46] was used with modifications. Plastic tape pieces, 20 mm long, were used as spacers between coverslips. A 7-mm square in the middle of the tape was cut out to create a chamber, and 3 μL of overnight culture mixed with 7 μL of fresh LB (1 mM IPTG) was placed in the chamber. This experiment was visualized using a confocal laser scanning microscope (FV3000RS, Olympus) with a 100×

objective (UPLAPO100XOHR, Olympus). Images were recorded at 5 frames per second for 20 seconds. For image analysis, cell edges in all images were highlighted by replacing each pixel with the neighborhood variance using the Variance tool in ImageJ [47] (https://imagej.net/ij/). Single-cell motility was analyzed using the TrackMate tool [48] (https://imagej.net/plugins/trackmate/) and the Simple LAC tracker was used for tracking. Track times (between the first spot of the track in time and the last spot of the track in time) of each cell were measured, and cells with a track time longer than 5.16 seconds were extracted to measure the following parameters: mean straight line speed (defined as the net displacement [Track displacement feature] divided by the track total time). To unify the number of cells used for statistical analyses, 80 cells of each strain were randomly selected. Histograms and normal distribution curves were created, and statistical significances were determined by Kolmogorov-Smirnov tests using Microsoft Excel for Mac Version 16.54. Representative video images are presented in (S1–S4 Videos).

## Western blot analysis

A previously described method [49] was used with modifications for fractionation of the membrane and cytoplasmic fractions. *B. subtilis* overnight cultures were washed twice with SMH (0.5 M Sucrose, 20 mM $MgCl_2$, 20 mM HEPES-NaOH pH 7.3) at room temperature. The cells were resuspended in 1/5 the volume of SMH and treated with lysozyme (10 mg/mL). The protoplasts were collected by centrifugation and flash-frozen in liquid nitrogen. Thawed protoplasts were dissolved in 500 μL of Buffer H (20 mM HEPES pH 7.3, 200 mM NaCl, 1 mM DTT, 1 tablet/50 mL protease inhibitor (Complete ULTRA, Roche)). The cell suspension was treated with 2 mM $MgCl_2$, 2 mM $CaCl_2$, DNaseI (0.02 U/μL), and RNaseA (40 μg/mL) at 4°C overnight and used as whole cell fraction. Debris was removed from the whole cell fraction by centrifugation at $6,000 \times g$ for 10 minutes at 4°C [50], and the supernatant was centrifuged at $100,000 \times g$ for 1 hour at 4°C. The pellet was dissolved in 100 μL of Buffer H to make the membrane fraction, and the supernatant was used as the cytosol fraction.

FLAG-tagged MotB was detected according to a previously described method [45]. Sodium dodecyl sulfate (SDS) sample buffer (final 1x) was added to each protein sample and boiled for 5 min. The samples were electrophoresed in a 12.5% SDS-polyacrylamide gel and the proteins were blotted to a 0.45-μm PVDF membrane (Millipore). An anti-DYKDDDDK (anti-FLAG) antibody (Fujifilm, Tokyo, Japan) diluted 1:5,000 in 1x TBST was used as the primary antibody. Horseradish peroxidase RP-conjugated anti-mouse IgG (Promega, Japan) diluted 1:5,000 in 1× TBST was used as the secondary antibody. Signals were detected using an Image-Quant LAS 4000 (Fujifilm, Tokyo, Japan).

## Blue native polyacrylamide gel electrophoresis

The membrane fractions of bacterial strains were treated with 0.5% n-dodesyl-β-D-maltoside and electrophoresed in a 4–20% gradient polyacrylamide gel in a EzRun BlueNative buffer (ATTO corp., Tokyo, Japan). The proteins in the gel were transferred to a 0.45-μm PVDF membrane (Millipore) using EzFastBlot HMW buffer (ATTO corp) and subjected to western blot analysis.

## Statistics

The statistical analyses were performed using Prism 9 software (GraphPad Software) unless otherwise specified. Relevant information on the statistical tests is provided in the captions of each figure.

## Supporting information

**S1 Fig. Growth curve of the MotB-overexpressed strain.** 50 μL of an $OD_{600}$-adjusted overnight culture of the vector-transformed strain (WT/EV) and MotB-overexpressed strain (WT/$motB_{OE}$) were aerobically cultured in 5 mL of LB containing 1 mM IPTG at 37˚C. Data shown are the means ± standard errors from four independent experiments.
(TIF)

**S2 Fig. Chemical structural formula of kanamycin and gentamicin.** Chemical structures were constructed by Marvin Sketch of ChemAxon (16.11.21.0).
(TIF)

**S1 Raw images. Original uncropped images for western blots in Fig 4B.**
(TIF)

**S2 Raw images. Original uncropped images for SDS polyacrylamide gel electrophoresis and western blots in Fig 5B.**
(TIF)

**S3 Raw images. Original uncropped images for western blots in Fig 6D and 6E and a marker in blue native polyacrylamide gel electrophoresis in Fig 6E.**
(TIF)

**S1 Video. Video data of swimming and cell tracks of WT/EV.** Swimming of the vector-transformed strain (WT/EV) captured under a bright field microscope (left) and detected cells and trajectories derived from automated tracking using TrackMate (right). Scale bar, 10 μm.
(MOV)

**S2 Video. Video data of swimming and cell tracks of WT/$motB_{OE}$.** Swimming of MotB-overexpressed strain (WT/$motB_{OE}$) captured under a bright field microscope (left) and detected cells and trajectories derived from automated tracking using TrackMate (right). Scale bar, 10 μm.
(MOV)

**S3 Video. Video data of swimming and cell tracks of Δ*motB*/EV.** Swimming of *motB*-deletion strain (Δ*motB*/EV) captured under a bright field microscope (left) and detected cells and trajectories derived from automated tracking using TrackMate (right). Scale bar, 10 μm.
(MOV)

**S4 Video. Video data of swimming and cell tracks of Δ*motB*/*motB*$_{OE}$.** Swimming of *motB*-deletion strain overexpressing MotB (Δ*motB*/*motB*$_{OE}$) captured under a bright field microscope (left) and detected cells and trajectories derived from automated tracking using TrackMate (right). Scale bar, 10 μm.
(MOV)

## Acknowledgments

We thank the National BioResource Project B. subtilis (National Institute of Genetics, Japan) for providing the BKE library and the Bacillus Genetic Stock Center for providing *B. subtilis* 168 and plasmids.

## Author Contributions

**Conceptualization:** Mio Uneme, Chikara Kaito.

**Data curation:** Mio Uneme, Kazuya Ishikawa, Kazuyuki Furuta, Atsuko Yamashita.

**Formal analysis:** Mio Uneme, Kazuya Ishikawa, Kazuyuki Furuta, Atsuko Yamashita.

**Funding acquisition:** Kazuya Ishikawa, Kazuyuki Furuta, Chikara Kaito.

**Investigation:** Mio Uneme, Atsuko Yamashita.

**Methodology:** Atsuko Yamashita.

**Supervision:** Chikara Kaito.

**Writing – original draft:** Mio Uneme.

**Writing – review & editing:** Mio Uneme, Kazuya Ishikawa, Kazuyuki Furuta, Atsuko Yamashita, Chikara Kaito.

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
