## [Decision Letter · Decision Letter 0]

20 Nov 2023

PONE-D-23-33472

Overexpression of the flagellar motor protein MotB sensitizes Bacillus subtilis to aminoglycosides in a motility-independent manner

PLOS ONE

Dear Dr. Kaito,

Thank you for submitting your manuscript to PLOS ONE. After careful consideration, we feel that it has merit but does not fully meet PLOS ONE’s publication criteria as it currently stands. Therefore, we invite you to submit a revised version of the manuscript that addresses the points raised during the review process.

The reviewers highlight major concerns about potential physiological complications arising from MotA/MotB protein overexpression and question the relevance of findings in a biological context. Differential expression levels of different stator types and the lack of systematic testing of different antibiotic concentrations hinder drawing reliable conclusions. Reviewer 2 raises a key point about protein misfolding potentially influencing observations. The revised manuscript should address these concerns and alternate possibilities through additional experiments and textual modifications.    

We look forward to receiving your revised manuscript.

Kind regards,

Pushkar P Lele

Academic Editor

PLOS ONE

Journal Requirements:

Reviewers' comments:

Reviewer's Responses to Questions

**Comments to the Author**

1. Is the manuscript technically sound, and do the data support the conclusions?

Reviewer #1: Yes

Reviewer #2: No

2. Has the statistical analysis been performed appropriately and rigorously? 

Reviewer #1: Yes

Reviewer #2: N/A

3. Have the authors made all data underlying the findings in their manuscript fully available?

Reviewer #1: Yes

Reviewer #2: Yes

4. Is the manuscript presented in an intelligible fashion and written in standard English?

Reviewer #1: Yes

Reviewer #2: Yes

5. Review Comments to the Author

Reviewer #1: This manuscript is admirable for its clarity and conciseness. I have absolutely no problems with it except as noted below with regard to Figure 5 and its interpretation and some suggestions for improving the figures. I also have some suggestions for further work, but I do not think they need to be incorporated into this preliminary report.

1) Questions about Figure 5A.

a) In the middle panel, the photographs do not seem consistent with the statistical evaluation. Overexpressed

MotB(D24E) looks no worse than EV, and overexpressed MotB(D24A) looks 10X better than either EV or MotB(D24E).

b) In the bottom panel. MotB(D24A) looks much worse than EV.

c) In the bottom panel, there is something wirtten that is illegible for the comparison of EV and MotB(D24E). This is

also true in Figure 1B for gentamycin.

d) Any idea why MotBD24A seems to be more abundant than WT MotB or MotBD24E?

2) General observation about the figures. The asterisks used to indicate P values are so small and blurred that they look

the same as the data points. Make them bigger/clearer.

3) The identity of SigP and AtpB should be made clear in the description of Figure 5B.

4) It is not clear from Figure 5A how important it is to have a D or an E at residue 24. I think the conclusion should be

tentative. It is also not clear why MotS does not have the same effect as MotB. Doesn't it also have an Asp residue at

the same relative position as D24 of MotB? I think the difference between proton-driven and sodium-ion-driven motility

relies on the MotA/MotP component. Was MotS expressed at the same level as MotB? It would seem that ion conduction

is not critical, so why would MotB and MotS be different?

5) On line 65, "played" should be "plays."

6) I think it would be useful to show the structures of kanamycin and gentamycin, perhaps in the Discussion, to highlight

why they might behave differently than the other antibiotics.

7) Suggestions for further work. I think they can be addressed as future investigations in the Discussion.

a) Does MotB have to associate with the peptidoglycan to have this effect? My guess would be no, as until MotB

associates with the flagellar motor, it probably does not extend its periplasmic domain to interact with the PG. However,

it should be easy enough to make MotB constructions that cannot interact with the PG.

b) Assuming that B. subtilis MotB also has a plug region, does deleting it affect the kanamycin/gentamycin

susceptibility either in the presence or absence of MotA.

c) To what extent does MotB overexpression disrupt the pmf? The diameter of a colony spreading in soft agar is a very

qualitative assay for motility, and even substantial reductions in the pmf might not affect the colony diameter very

much. This is a critical point, as the authors postulate that it is a decrease in the pmf that causes the increased

susceptibility to kanamycin and gentamycin. Therefore, better quantitative measures of flagellar performance and pmf

will be necessary to draw definitive conclusions. I think this question should be addressed a bit more fully in the

Discussion and alternative explanations, if any suggest themselves, should be considered.

Reviewer #2: Uneme and authors propose to test the effects of flagellar proteins on susceptibility to aminoglycosides. To test this, they overexpress flagellar proteins MotA and MotB individually and together in B. subtilis and examine the effects on motility and sensitivity to aminoglycosides. Overexpression of MotB leads to an increase in aminoglycoside sensitivity but does not affect sensitivity to the other classes of antibiotics tested. This sensitivity does not require overexpression of MotA, its usual binding partner. Over-expression of MotA does not alter sensitivity to aminoglycosides. Mutations in the D24A proton binding site of MotB eliminates the sensitivity observed when overexpressed. Overexpression of MotB doesn’t affect motility-dependent spreading on a soft-agar plate. The authors conclude that MotB has a novel function by increasing the susceptibility to aminoglycosides.

My primary concern is that the authors are overexpressing MotB at levels that are far an above what would normally be observed in the cell, and in doing so, are measuring phenotypes that aren’t relevant in a biological context. Therefore, the conclusion that MotB has a novel function is not supported.

Instead of a novel function, it seems more likely that MotB is misfolding because it doesn’t have its binding partner, MotA, or because protein expression is too high for the membrane to adequately support. This could lead to slightly leaky membranes that are sufficient to alter the membrane potential but still support cell division and growth. The D24A mutation in the proton binding site could be support for MotB folding properly, but it’s also likely that D24A mutation leads to a different misfolding/mis-localization since you’re eliminating a charged amino acid in the hydrophobic/transmembrane portion of the protein.

Is the doubling time altered upon overexpression of MotB?

A p-value of 0.0549 isn’t significant so the authors cannot conclude that there is a “tendency of decreased growth compared to the vector-transformed strain.” (line 76)

The statement that overexpression of MotB is motility-independent needs to be qualified. The soft-agar assay is a single time point assay that requires motility but is not sufficient to say that there is no difference in motility. The rate of spreading depends on the motility of single cells but also the rate of nutrient depletion in the agar and rate of cell division. If nutrient depletion is rate limiting, then differences in motility may not be observed. Single-cell motility assays are needed to conclude that motility isn’t impacted by overexpression of MotB.

How are single cells counted in the spot assays in Figure 2 when no individual colonies are observed, e.g., wt motB-FLAG(OE) with kanamycin or gentamicin? Where does the data come from in the bottom half of the figure? The authors state that there is “decreased growth” (line 109), but isn’t it a decreased survival since you’re reducing the number of colonies?

6. PLOS authors have the option to publish the peer review history of their article (what does this mean?). If published, this will include your full peer review and any attached files.

Reviewer #1: No

Reviewer #2: No

---

## [Author Response · Author response to Decision Letter 0]

27 Jan 2024

Our responses to the editor's and reviewers' comments

Editor’s comment:

The reviewers highlight major concerns about potential physiological complications arising from MotA/MotB protein overexpression and question the relevance of findings in a biological context. Differential expression levels of different stator types and the lack of systematic testing of different antibiotic concentrations hinder drawing reliable conclusions. Reviewer 2 raises a key point about protein misfolding potentially influencing observations. The revised manuscript should address these concerns and alternate possibilities through additional experiments and textual modifications. 

Response: We thank the editor for summarizing the reviewers' comments. We agree with the possibility that overexpressed MotB has a misfolding conformation. Since we do not assume that overexpressed MotB mimics a physiological state of MotB, we have revised the manuscript to not mention the function of MotB in a biological context.

 To examine the folding of overexpressed MotB, we performed new experiments using blue native polyacrylamide gel electrophoresis. We found that overexpressed MotB exists as a homodimer or several multicomplexes. The amount of homodimer did not differ between overexpressed MotB and non-overexpressed MotB, whereas the amount of multicomplexes was greater in overexpressed MotB than in non-overexpressed MotB. The results suggest that the multicomplexes of MotB may contribute to the aminoglycoside sensitivity. We have added new data in Figure 6 and explained the results on pages 13–14, lines 229–251 in the revised manuscript.

 According to the editor's comment on antibiotic concentration, we performed a new experiment to examine MIC values. We have presented the data in Table 1 and added an explanation on page 8, lines 140–143 of the revised manuscript. 

Reviewers' comments:

Reviewer #1: This manuscript is admirable for its clarity and conciseness. I have absolutely no problems with it except as noted below with regard to Figure 5 and its interpretation and some suggestions for improving the figures. I also have some suggestions for further work, but I do not think they need to be incorporated into this preliminary report.

Response: We are grateful for your positive and encouraging comments. Our responses related to Figure 5 and your suggestions are described below.

1) Questions about Figure 5A.

a) In the middle panel, the photographs do not seem consistent with the statistical evaluation. Overexpressed MotB(D24E) looks no worse than EV, and overexpressed MotB(D24A) looks 10X better than either EV or MotB(D24E). 

Response: We agree with the reviewer's point. Accordingly, we have added the photographs which represent the average.

b) In the bottom panel. MotB(D24A) looks much worse than EV.

Response: The survival of D24A was slightly worse than that of EV in the presence of aminoglycosides, although the difference was not statistically significant. We have corrected the explanation to “did not show significantly decreased survival” (line 197) in the revised manuscript.

c) In the bottom panel, there is something written that is illegible for the comparison of EV and MotB(D24E). This is also true in Figure 1B for gentamycin.

Response: We believe that what the reviewer pointed out is the described P-value. According to the other reviewer's suggestion, we have deleted this P-value in Figures 1B and 5A. 

d) Any idea why MotBD24A seems to be more abundant than WT MotB or MotBD24E?

Response: We speculate that the reason is due to the structural stabilization of the MotA/MotB complex by alanine substitution, as a previous report suggested that the amino acid substitution affects the structure of the MotA/MotB complex (Kojima S, Blair DF. Biochemistry. 40, 13041–50 (2001)). This explanation has been added on page 12, lines 211–214 in the revised manuscript.

2) General observation about the figures. The asterisks used to indicate P values are so small and blurred that they look the same as the data points. Make them bigger/clearer.

Response: Thank you for this suggestion. We have enlarged the asterisks in the figures.

3) The identity of SigB and AtpB should be made clear in the description of Figure 5B.

Response: According to the reviewer's suggestion, we have included the following explanation: "FoF1-ATP synthase subunit a (AtpB; [28]) and an RNA polymerase sigma factor (SigB; [29,30]) were used as marker proteins for the membrane fraction and the cytosolic fraction, respectively” on page 12, lines 202–204 in the revised manuscript.

4) It is not clear from Figure 5A how important it is to have a D or an E at residue 24. I think the conclusion should be tentative. It is also not clear why MotS does not have the same effect as MotB. Doesn't it also have an Asp residue at the same relative position as D24 of MotB? I think the difference between proton-driven and sodium-ion-driven motility relies on the MotA/MotP component. Was MotS expressed at the same level as MotB? It would seem that ion conduction is not critical, so why would MotB and MotS be different?

Response: According to the reviewer's comment that MotS may not be expressed, we performed a new experiment to measure the amount of overexpressed MotS. We confirmed that MotS was expressed. We have added the result in Figure 4B and explained the result on page 10, lines 170–173 in the revised manuscript.

 The Asp 24 residue is conserved between MotB and MotS. Based on the report that the ion selectivity was retained when MotA and MotP were interchanged (Ito M. et al., J Mol Biol, 352, 396–408 (2005)), the ion binding sites of MotB and MotS are assumed to be important for discrimination between protons or sodium ions. We have explained these points on page 11, lines 189–190 in the revised manuscript.

5) On line 65, "played" should be "plays."

Response: According to the reviewer's suggestion, we have corrected the word in the revised manuscript.

6) I think it would be useful to show the structures of kanamycin and gentamycin, perhaps in the Discussion, to highlight why they might behave differently than the other antibiotics.

Response: According to the reviewer's suggestion, we have added a discussion on the structural differences between kanamycin and gentamycin on page 16, lines 284–287 in the revised manuscript. We have also added the chemical structural formulas of kanamycin and gentamicin in S2 Fig.

7) Suggestions for further work. I think they can be addressed as future investigations in the Discussion.

a) Does MotB have to associate with the peptidoglycan to have this effect? My guess would be no, as until MotB associates with the flagellar motor, it probably does not extend its periplasmic domain to interact with the PG. However, it should be easy enough to make MotB constructions that cannot interact with the PG.

Response: We are grateful for your suggestions for future investigation. We agree with the reviewer's assumption that overexpressed MotB does not interact with PG, as MotB interacts with PG when MotB forms a complex with MotA and the flagella motor. We have added the future investigation suggestions to examine whether the PG binding domain is required for the aminoglycoside sensitivity on page 17–18, lines 307–310 in the revised manuscript. 

b) Assuming that B. subtilis MotB also has a plug region, does deleting it affect the kanamycin/gentamycin susceptibility either in the presence or absence of MotA.

Response: Thank you for this suggestion. As the reviewer pointed out, the plug region of B. subtilis MotB has not been characterized. We have added the plan to investigate the involvement of the plug region of MotB for the aminoglycoside-sensitive phenotype on page 17–18, lines 307–310 in the revised manuscript. 

c) To what extent does MotB overexpression disrupt the pmf? The diameter of a colony spreading in soft agar is a very qualitative assay for motility, and even substantial reductions in the pmf might not affect the colony diameter very much. This is a critical point, as the authors postulate that it is a decrease in the pmf that causes the increased susceptibility to kanamycin and gentamycin. Therefore, better quantitative measures of flagellar performance and pmf will be necessary to draw definitive conclusions. I think this question should be addressed a bit more fully in the Discussion and alternative explanations, if any suggest themselves, should be considered.

Response: According to the reviewer's suggestion, we performed a quantitative measurement of flagella performance using a single-cell motility assay. We found that single-cell motility was not affected by MotB overexpression. The result suggests that MotB overexpression causes aminoglycoside-sensitive phenotype without affecting motility and possibly without affecting the PMF. We have added new data in Figure 1D and discussed the involvement of PMF on page 17, lines 300–303 in the revised manuscript. 

Reviewer #2: Uneme and authors propose to test the effects of flagellar proteins on susceptibility to aminoglycosides. To test this, they overexpress flagellar proteins MotA and MotB individually and together in B. subtilis and examine the effects on motility and sensitivity to aminoglycosides. Overexpression of MotB leads to an increase in aminoglycoside sensitivity but does not affect sensitivity to the other classes of antibiotics tested. This sensitivity does not require overexpression of MotA, its usual binding partner. Over-expression of MotA does not alter sensitivity to aminoglycosides. Mutations in the D24A proton binding site of MotB eliminates the sensitivity observed when overexpressed. Overexpression of MotB doesn’t affect motility-dependent spreading on a soft-agar plate. The authors conclude that MotB has a novel function by increasing the susceptibility to aminoglycosides.

Response: We thank the reviewer for the clear summary of our findings.

My primary concern is that the authors are overexpressing MotB at levels that are far an above what would normally be observed in the cell, and in doing so, are measuring phenotypes that aren’t relevant in a biological context. Therefore, the conclusion that MotB has a novel function is not supported.

Response: We thank the reviewer for this comment and agree that overexpressed MotB cannot mimic the physiological condition. According to the reviewer's suggestion, we have deleted the expression "a novel function" and the related explanation in the revised manuscript. 

Instead of a novel function, it seems more likely that MotB is misfolding because it doesn’t have its binding partner, MotA, or because protein expression is too high for the membrane to adequately support. This could lead to slightly leaky membranes that are sufficient to alter the membrane potential but still support cell division and growth. The D24A mutation in the proton binding site could be support for MotB folding properly, but it’s also likely that D24A mutation leads to a different misfolding/mis-localization since you’re eliminating a charged amino acid in the hydrophobic/transmembrane portion of the protein.

Response: We thank the reviewer for this comment and agree that overexpressed MotB could exist as misfolding proteins. To understand the folding of overexpressed MotB or complex formation by overexpressed MotB, we performed new experiments using blue native PAGE. We found that overexpressed MotB exists as a homodimer or several multicomplexes. The amount of homodimer did not differ between overexpressed MotB and non-overexpressed MotB, whereas the amount of multicomplexes was greater in overexpressed MotB than in non-overexpressed MotB. The results suggest that the multicomplexes of MotB may contribute to the aminoglycoside sensitivity. As the reviewer pointed out, such multicomplexes might damage the membrane. We have added new data in Figure 6 and explained the results on pages 13–14, lines 229–251 in the revised manuscript.

Is the doubling time altered upon overexpression of MotB?

Response: We thank the reviewer for this question. The overexpression of MotB did not affect the doubling time. The result suggests that overexpressed MotB does not cause a severe leakage of the membrane. We explained the result on pages 4–5, lines 70–72, and added the data in S1 Fig in the revised manuscript. 

A p-value of 0.0549 isn’t significant so the authors cannot conclude that there is a “tendency of decreased growth compared to the vector-transformed strain.” (line 76)

Response: Thank you for this comment. We have deleted the sentence in the revised manuscript.

The statement that overexpression of MotB is motility-independent needs to be qualified. The soft-agar assay is a single time point assay that requires motility but is not sufficient to say that there is no difference in motility. The rate of spreading depends on the motility of single cells but also the rate of nutrient depletion in the agar and rate of cell division. If nutrient depletion is rate limiting, then differences in motility may not be observed. Single-cell motility assays are needed to conclude that motility isn’t impacted by overexpression of MotB.

Response: We thank the reviewer for raising this important point. According to the reviewer's suggestion, we performed a quantitative measurement of flagella performance using a single-cell motility assay. We found that single-cell motility was not affected by MotB overexpression. The result suggests that MotB overexpression causes aminoglycoside-sensitive phenotype without affecting motility. We have added new data in Figure 1D and explained the result on pages 5–6, lines 82–89 in the revised manuscript. 

How are single cells counted in the spot assays in Figure 2 when no individual colonies are observed, e.g., wt motB-FLAG(OE) with kanamycin or gentamicin? Where does the data come from in the bottom half of the figure? The authors state that there is “decreased growth” (line 109), but isn’t it a decreased survival since you’re reducing the number of colonies?

Response: We thank the reviewer for this comment and agree that our assay measures bacterial survival. We have corrected the word "growth" to "survival" in the revised manuscript. 

 We have also explained how we counted the bacterial colonies on page 22, lines 366–370 in the revised manuscript as follows. 

i) When 5 or more colonies were observed in the most diluted spot, we counted the number of colonies.

ii) When 4 or fewer colonies were observed in the most diluted spot, we counted the number of colonies in the next most diluted spot.

iii) When the number of colonies could not be counted due to crowded colonies, we considered the number of colonies to be 20.

iv) When the colonies were observed only at the edge of the spot, like a circle, we considered the number of colonies to be 10.

---

## [Decision Letter · Decision Letter 1]

4 Mar 2024

Overexpression of the flagellar motor protein MotB sensitizes Bacillus subtilis to aminoglycosides in a motility-independent manner

PONE-D-23-33472R1

Dear Dr. Kaito,

We’re pleased to inform you that your manuscript has been judged scientifically suitable for publication and will be formally accepted for publication once it meets all outstanding technical requirements.

Kind regards,

Pushkar P Lele

Academic Editor

PLOS ONE

Additional Editor Comments (optional):

Reviewers' comments:

Reviewer's Responses to Questions

**Comments to the Author**

1. If the authors have adequately addressed your comments raised in a previous round of review and you feel that this manuscript is now acceptable for publication, you may indicate that here to bypass the “Comments to the Author” section, enter your conflict of interest statement in the “Confidential to Editor” section, and submit your "Accept" recommendation.

Reviewer #2: All comments have been addressed

2. Is the manuscript technically sound, and do the data support the conclusions?

Reviewer #2: Yes

3. Has the statistical analysis been performed appropriately and rigorously? 

Reviewer #2: Yes

4. Have the authors made all data underlying the findings in their manuscript fully available?

Reviewer #2: Yes

5. Is the manuscript presented in an intelligible fashion and written in standard English?

Reviewer #2: Yes

6. Review Comments to the Author

Reviewer #2: The authors' revised manuscript addressed all of my concerns. The growth curve and single-cell motility assays together demonstrate that the cells aren't significantly stressed with an altered PMF. The native gel is a valuable addition the manuscript because it reveals the distribution of motB in different complexes (at least under the conditions of the gel, which has detergent present). The authors hypothesize that some motB are in homo-dimers, some might be in MotAB complexes, and some are in higher molecular weight complexes, perhaps with other flagellar proteins or as very large homo-aggregates.

It's still not clear how an increased motB leads to an increased kanamycin susceptibility, which is the primary limitation of the study. Is motB acting by itself because there is not enough motA to form a complex? Is motB acting with another protein that it natively associates with but only when there is excess motB? Or is motB associating with another protein that it never binds but can at these high expression levels? In other words, is overexpressing motB revealing a new, perhaps rare, function for motB or is it an overexpression artifact?

7. PLOS authors have the option to publish the peer review history of their article (what does this mean?). If published, this will include your full peer review and any attached files.

Reviewer #2: No
